# Effects of sub-word segmentation
# on performance of transformer language models

**Jue Hou,**[*†] **Anisia Katinskaia,**[*†] **Anh-Duc Vu,**[*†] **Roman Yangarber**[†]
[*] Department of Computer Science
[†] Department of Digital Humanities
University of Helsinki, Finland
`first.last@helsinki.fi`

## Abstract

Language modeling is a fundamental task in natural language processing, which has been thoroughly explored with various architectures and hyperparameters. However, few studies focus on the effect of *sub-word segmentation* on the performance of language models (LMs). In this paper, we compare GPT and BERT models trained with the statistical segmentation algorithm BPE vs. two unsupervised algorithms for morphological segmentation—Morfessor and StateMorph. We train the models for several languages—including ones with very rich morphology—and compare their performance with different segmentation algorithms, vocabulary sizes, and model sizes. The results show that training with morphological segmentation allows the LMs to: **1.** achieve *lower perplexity*, **2.** converge *more efficiently* in terms of training time, and **3.** achieve equivalent or better evaluation scores on *downstream* tasks. Lastly, we show **4.** that LMs of smaller size using morphological segmentation can perform comparably to models of larger size trained with BPE—both in terms of (1) perplexity and (3) scores on downstream tasks. Points 2 and 4 impact on **sustainability**, since they reduce the model cost; and while 2 reduces cost only in the training phase, 4 does so also in the inference phase.

## 1 Introduction

A key component required to train language models is the tokenizer. One "basic" naïve approach is *word-based* tokenization—splitting the input into words, and assembling a training sequence as the list of word tokens. The problem with this approach is out-of-vocabulary tokens (OOV) during inference—i.e., tokens not seen during training. One solution is to pick a set of most frequent words, and replace all other words with a special [UNK] token (Bahdanau et al., 2015; Sutskever et al., 2014). This works well only when the unknown vocabulary is small (Cho et al., 2014; Bahdanau et al.,

2015). Another solution is to use an extensive dictionary (Jean et al., 2015; Luong et al., 2015). This approach is problematic for morphologically-rich languages, such as Finnish or Russian. The word-based approach results in a huge vocabulary, and still very many OOV tokens, due to the possibly rare morphological variants (Sennrich et al., 2016).

At the other extreme is character-based tokenization—treating each character as a separate token; but this leads to long sequences, and non-meaningful tokens, which inevitably increase the computational complexity (Libovický et al., 2022). Hence, the most common approach is in the middle—"sub-word" tokenization. A *segmenter* is an algorithm that splits words into sub-word units, between word- and character-based tokenization.

The most common choice for a segmenter is the "Byte-Pair Encoding" (BPE) algorithm, originally designed for simple data compression (Gage, 1994). Currently most state-of-the-art neural language models, such as GPT (Radford et al., 2019) and BERT (Devlin et al., 2018), use it as the segmenter. The idea behind BPE and its variants is to compute the frequencies of all segments, to iteratively merge the most frequent consecutive pair of segments into one segment, and add it to the vocabulary of tokens. This yields a compression of the data, and also a way to represent the data using a finite vocabulary of segment tokens—much smaller than by using entire word tokens.

BPE works well in training language models, as it reduces the size of the vocabulary while still preserving some frequent words in the language as tokens. However, BPE has several problems. It is a greedy algorithm, and gives a crude approximation to the "true" linguistic structure of words, that is, *morphemes*. A morpheme is a linguistically meaningful segment of a word—it carries either semantic or syntactic content, and cannot be split further into smaller segments. Also, morpheme retains its meaning in *different* contexts.

The goal of this work is to determine whether we can improve the performance of language models by using segmenters that are more sophisticated than BPE—linguistically motivated and morphologically informed. For example, BPE may decide to segment the English word *baking* into two frequent segments: *ba·king*. However, we may consider *bak·ing* a "better" segmentation: root morpheme *bak* and suffix morpheme *ing*. It stands to reason that "knowing" about morphology—having more *meaningful* segments in its vocabulary—should help the LM predict data better. To be fair, the LM might be able to "recover" from the naïve segmentation, and still achieve excellent performance on various tasks, but it will have to do so *at a higher cost*!—in terms of more parameters and longer training—which degrades sustainability.

Which segmentation algorithm yields a *better* language model? We explore the "goodness" of a LM through four research questions:

• **RQ1**: compared to BPE, does morphological segmentation help LMs achieve lower perplexity?

• **RQ2**: compared to BPE, does morphological segmentation help LMs converge faster?

• **RQ3**: compared to BPE, does morphological segmentation allow LMs to achieve similar (or better) performance on downstream tasks?

• **RQ4**: compared to BPE, does morphological segmentation allow us to reduce model size without compromising model performance?

The paper is organized as follows: Section 2 briefly reviews prior work on language modeling with morphological features, and the segmentation methods. Section 3 introduces the language models and the downstream tasks used for evaluation. Section 4 presents the results of our experiments. Section 5 concludes and discusses future work.

## 2 Prior Work

Sub-word tokenization is a well-studied topic in natural language processing. Several approaches are proposed to segment words into sub-word units, (Batsuren et al., 2022; Peters and Martins, 2022; Minixhofer et al., 2023). However, few papers have studied the effect of sub-word segmentation on the performance of neural language models.

Hofmann et al. (2021) discuss how sub-word segmentation affects BERT's interpretation of complex words. They conduct a series of semantic probing tasks and show that applying morphological segmentation is beneficial for language models.

Park et al. (2021) trained models with several segmentation algorithms, including BPE and Morfessor (Creutz and Lagus, 2002), on a corpus of Bible verses in 92 languages. They evaluate the models according to *surprisal* (negative log-likelihood) per verse. Their results show that Morfessor segmentation yields much lower surprisal per verse than character or BPE segmentation for the overwhelming majority of the tested languages.

Bostrom and Durrett (2020) compare the output of BPE and Unigram tokenization (Kudo, 2018) on English and Japanese. Comparing with gold-standard morpheme boundaries, they found that Unigram tokenization segments words more closely to morphological references, while BPE greedily merges sub-words according to their frequency, even if the sub-word pair is not semantically meaningful. They experimented with evaluating language models on downstream tasks, and pre-trained models from scratch with various segmentation methods. They found that Unigram tokenization outperforms BPE on both English and Japanese.

Toraman et al. (2022) conduct a comprehensive study to evaluate how tokenization affects the performance of Turkish LMs. Besides BPE and its variant WordPiece (Wu et al., 2016), they apply a morphological analysis tool for Turkish, and use it as sub-word segmenter. Similarly to (Bostrom and Durrett, 2020), they conducted experiments by evaluating six downstream LM tasks. Although BPE and WordPiece achieve the best performance, LM with morphology-based segmentation achieve comparable results. They also point out that LM with a bigger vocabulary size can generally perform better. However, a large vocabulary increases the computational complexity.

### 2.1 Sub-word segmentation

We briefly introduce the three segmentation algorithms we use in this work. We leave out the mathematical details, and refer the reader to the original papers for further information.

**BPE:** is one of most popular sub-word segmentation algorithms (Gage, 1994). It is a greedy algorithm, that starts from a character-based "segmentation" (tokenization) of each word, then iteratively sorts all adjacent pairs of segments by frequency of co-occurrence, and *merges* the most frequent adjacent pair. The merged pair is added to the vocabulary, until the vocabulary reaches a pre-selected maximum size. At this point, BPE stops merging.

We use Google's *SentencePiece* implementation of BPE.

**Morfessor:** is an unsupervised algorithm for morphology induction (Creutz and Lagus, 2002). Based on the Minimum Description Length (MDL) principle (Rissanen, 1998; Grünwald, 2007), it favors lexicons with fewer and shorter sub-words. It learns the model parameters with MAP estimation using a two-part cost function: a prior cost and a corpus cost. The corpus cost encodes the probability of the corpus given the lexicon and segmentation.

The original Morfessor has been extended in several ways (Grönroos et al., 2020, 2014); however, the later implementations are not included in the official Python library distribution. Therefore, we use the Morfessor 2.0 implementation (Virpioja et al., 2013) of the baseline method. We emphasize that in the Morfessor baseline that we use, vocabulary size fixed, determined by the training (not customizable). Exploring other variants with configurable vocabulary sizes is left for future work.

**StateMorph:** is based on the MDL principle similarly to Morfessor, but is different in several respects (Nouri and Yangarber, 2017). It tries to model the morphological structure of the language using *states* in a finite-state network. Each state learns to emit segments ("morphs") of a certain kind, e.g., verb stems, noun suffixes, etc. It also uses a MDL-based two-part objective: the cost of the *complete* data—i.e., the *segmented* corpus—is the cost of the model, plus the cost of the data given the model. The cost of the model consists of three components: the cost of the morph lexicon, the cost of transitions between states, and the cost of emitting morphs from states. Unlike Morfessor, StateMorph uses simulated annealing to optimize the search, which makes learning much slower. We use the baseline implementation released with the original paper.[1] It decides on its own optimal lexicon size, according to the optimized objective.

In our experiments, we also use a variant of State-Morph segmentation, where we configure the desired lexicon size. Similarly to Morfessor baseline, we cannot control the size of StateMorph's lexicon during training; however, we can prune the lexicon *after* training, by simply dropping the least frequent morphs. The resulting segmenter will tend to segment a word with more frequent sub-words (even if the overall cost may be higher). But this allows us to compare language models under similar conditions: with same lexicon size. In the following, we denote normal StateMorph by **SM**, and StateMorph with a *pruned* lexicon by **SMp**.

## 3 Methods

To evaluate the impact of the segmentation on the language models, we conduct several experiments. For each LM—BERT and GPT, we perform four evaluations, corresponding to the four research questions. Each model is trained with the sub-word lexicon resulting from the three segmentation algorithms: BPE, Morfessor, and StateMorph.

### 3.1 Training the language models

RQ1 asks: which of the segmentation algorithms yields a better language model—in terms of *perplexity*? In the information-theoretic sense, this is the definitive measure of the model's "goodness": perplexity tells how well the model is able to predict the data. Thus, in theoretical terms, a model with lower perplexity is a better model.

We also keep track of how many steps each LM takes to converge, to answer RQ2: does a smarter segmentation help the LM learn faster.

Hyperparameters used in training follow the same settings as for $BERT_{base}$ in (Turc et al., 2019), except we use a smaller feed-forward/filter size to reduce the model size. We describe all settings and hyperparameters in detail in Appendix A.

We used a smaller instance size—256, half of what is typically used for BERT. This is due to limited computational resources and exceptionally large vocabulary size for some LMs. We were not able to train with a conventional batch size. As the instance size is smaller than regular and resources are limited, we would like to focus only on experimenting with the effect of segmentation and avoid spare resources on harder side-tasks such as next-sentence prediction. Therefore, we did not include it as a part of BERT pre-training. We aim for sentence-level LMs (rather than larger context). In future work, we can repeat this on bigger models properly, with more computational resources.

**Data:** the corpora used to train the language models are as follows. The Finnish corpus contains data from two major Finnish news agencies: Helsingin Sanomat (HS) and YLE.[2] The corpus contains around 17M instances. For Russian, we use the Taiga corpus (Shavrina and Shapovalova,

---

[1] http://nlp.cs.helsinki.fi/morpho

[2] MetaShare: Yle Finnish News Archive

2017). The corpus contains 66M instances. The English corpus comes mainly from the English Wikipedia dump from 2020-05-01.[3] We add our own news data, privately crawled from the Internet. The corpus contains 61M instances. For Turkish, we use the OSCAR corpus (Abadji et al., 2022). The corpus contains 20M instances. Each training instance is composed of 3 sentences.

**Pre-processing:** For each language, we train all segmentation algorithms with the same word list, extracted from the training corpus. We lowercase all words to assure that the segmentation of a given word is the same regardless of its case. However, the language models should handle mixed-case input. Additional technical details about the pre-training phase are given in Appendix A.1.

To make the comparison fair, we make sure that the experiments use the same vocabulary size. As mentioned above, we are not able to customize the lexicon size for Morfessor directly. To customize the lexicon size for StateMorph, we prune the lexicon by frequency of the morph. Therefore, we set the lexicon size for BPE manually—to match the lexicon size produced by Morfessor and State-Morph. We leave the exploration of the effect of StateMorph variants and Morfessor variants with adjustable lexicon size for future work.

We should clarify that we do not aim for optimal lexicon size in terms of LM pre-training or any of the downstream tasks; it is only to make the comparison fair. Searching for the optimal lexicon size would require exorbitant compute resources, and is likely highly language-specific. This is not our goal here. The sizes of the resulting lexicons are shown in Appendix A.

## 3.2 Model size

RQ4 asks whether a smarter segmentation will allow us to build a language model that has equivalent or better performance with smaller model size—measured in terms of the number of learned model parameters. This is a crucial question for *sustainability*, since the large language models consume vast amounts of computing resources—both in the training and in the inference phases.

We configure our small models similarly to $BERT_{medium}$ in (Turc et al., 2019); more details about hyperparameters are in Appendix A.

## 3.3 Downstream tasks

RQ3 asks whether we can confirm that language models with lower perplexity yield comparable (or better) performance on downstream tasks. Thus we also evaluate performance of the models on practical applications. We consider two types of tasks: classification vs. generative tasks.

**Finnish:** we use two topic classification tasks, and Part-of-Speech (PoS) tagging, which is a sequence-labeling task. Topic classification is based on two corpora: In-domain and Out-of-domain. For the In-domain task, we use the YLE corpus, same as for training the segmentation algorithms and LMs. We classify documents into four topics: Sport, Politics, Science, and Culture. For the Out-of-domain task, we use the dataset from Ylilauta[4] with topics: Sport, Politics, Science, and Food&Drink. We use the same instance size as in pre-training. For each corpus, we use 10k, 1k, and 1k for training, validation, and testing, respectively. The PoS tagging task is based on *Finnish-TDT* from UD: Universal Dependency (Nivre et al., 2020).

For the generative task, we use a paraphrase dataset from (Kanerva et al., 2021). We use 21k for training, 2.6k for validation, and 2.6k for testing.

**Russian:** For classification, we use topic classification based on the *Lenta.ru* corpus; part-of-speech tagging, based on *Russian-SynTagRus* from UD; and a linguistic acceptability (LA) task, using RuCoLa (Mikhailov et al., 2022), with GPT and BERT models. For the generative task, we use paraphrase generation with ru-paraphrase-NMT-Leipzig dataset.[5]

We explore PoS tagging only with BERT, as GPT is a left-to-right LM, which prevents the model's access to the "future" during training, while PoS tagging may require inference from the entire surrounding context, not only the left side. Therefore, GPT may not be suitable for PoS tagging.

We explore the paraphrase generation task only with GPT models, as we did not include next-sentence prediction in pre-training our BERT—and therefore it may not be suitable for generative tasks. We leave this for future work.

## 4 Experiments and results

This section presents a series of experiments that we conduct to address the research questions.

[3]HuggingFace Wikipedia datasets

[4]MetaShare: Ylilauta Corpus
[5]HuggingFace: ru-paraphrase-NMT-Leipzig

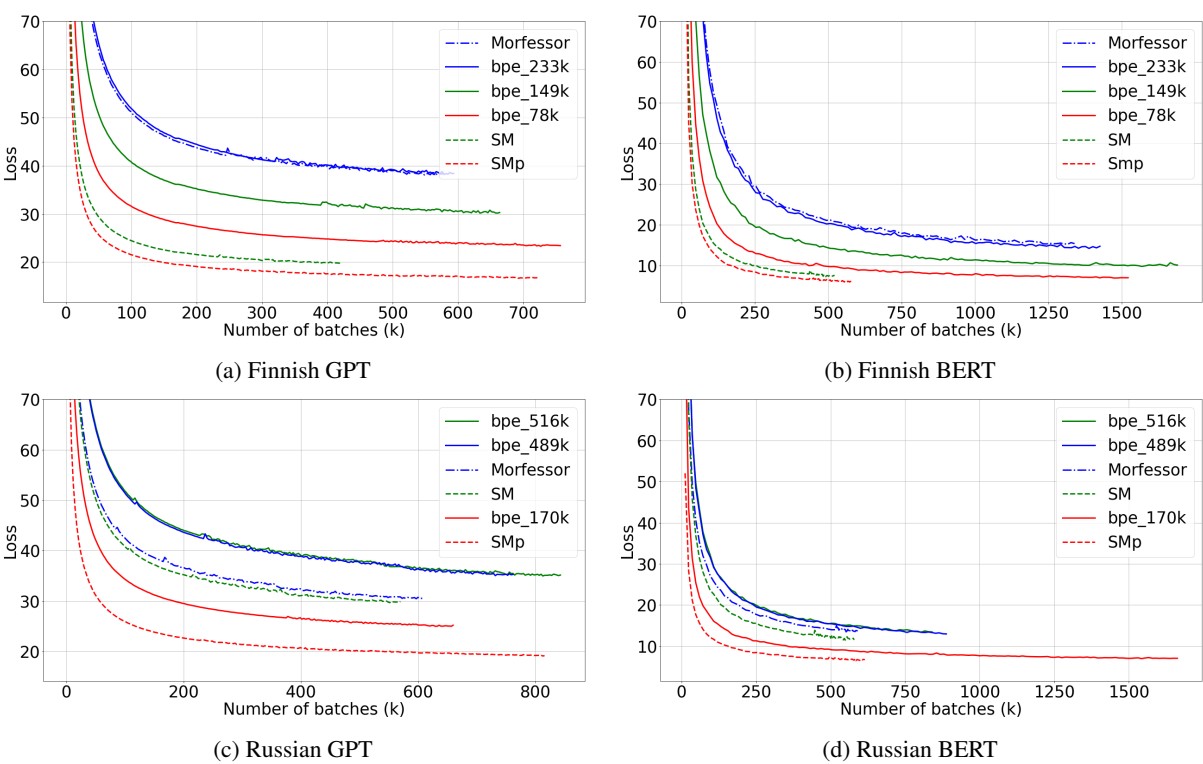

Figure 1: Learning curves for pre-training Finnish and Russian LMs with different segmentation methods.

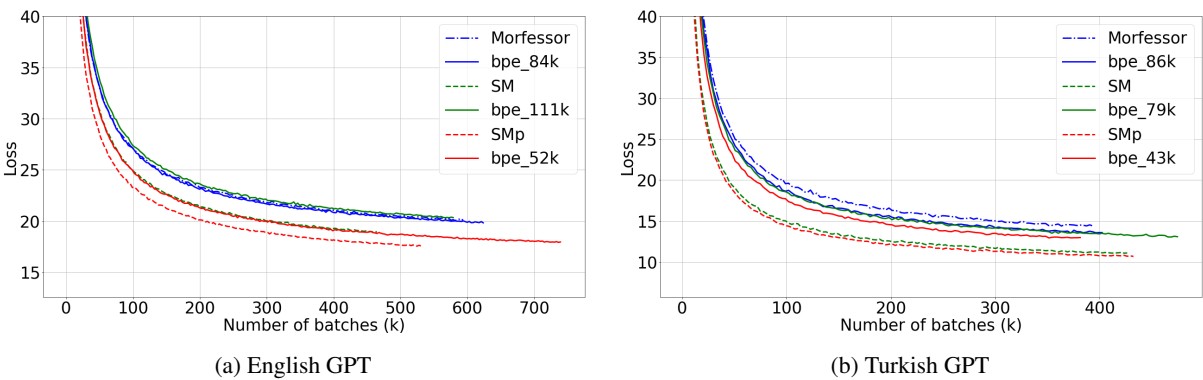

Figure 2: Learning curves for pre-training English and Turkish GPT with different segmentation methods.

## 4.1 Pre-Training with different segmentations

To explore the behavior of the different segmentation methods, we pretrain an array of LMs, GPT and BERT. We visualize the learning curves of Finnish and Russian models in Figure 1. Solid lines indicate BPE, lines with dashes indicate morphological segmentation; line colors code the vocabulary sizes. The same experiments for English and Turkish, GPT only, are shown in Figure 2.

We show the final perplexity, and the number of steps to reach convergence—Finnish and Russian in Table 1, English and Turkish in Table 2.

The curves and the Tables show that—with the same vocabulary size—for almost all models, mor-

phological segmentation yields lower perplexity. There are only a few exceptions: BERT with Morfessor (FI and RU) and GPT with Morfessor (EN and TR); but perplexity is very near to the BPE counterpart. This suggests models with morphological segmentation yield overwhelmingly better perplexity, compared to using BPE, with the same vocabulary size (RQ1).

Regarding the impact of segmentation on the learning progress: the number of steps to reach convergence (first column in both tables) is higher for most models with BPE, several times higher for some. This suggests that the models with morphological segmentation are also *more efficient* in

| | Segmenter | GPT steps (k) | GPT PPL | BERT steps (k) | BERT PPL |
|---|---|---|---|---|---|
| Finnish | bpe (233k) | 593 | 38.17 | 1426 | 14.18 |
| | Morfessor | 578 | 38.09 | 1339 | 15.20 |
| | bpe (149k) | 664 | 30.16 | 1690 | 9.75 |
| | SM | 419 | 19.71 | 521 | 7.23 |
| | bpe (78k) | 757 | 23.69 | 1522 | 6.89 |
| | SMp | 724 | **16.65** | 584 | **5.92** |
| Russian | bpe (488k) | 762 | 35.01 | 888 | 12.96 |
| | Morfessor | 609 | 30.43 | 591 | 13.43 |
| | bpe (516k) | 843 | 34.96 | 840 | 13.40 |
| | SM | 363 | 29.66 | 579 | 11.47 |
| | bpe (170k) | 660 | 24.97 | 1663 | 6.97 |
| | SMp | 816 | **19.08** | 615 | **6.47** |

Table 1: Pre-training results for Finnish and Russian LMs with different segmentation methods. Vocabulary size (in parentheses) applies to all models in each box.

| | Segmenter | GPT steps (k) | GPT PPL |
|---|---|---|---|
| English | bpe (151k) | 624 | 19.82 |
| | Morfessor | 594 | 20.05 |
| | bpe (177k) | 579 | 20.33 |
| | SM | 465 | 18.93 |
| | bpe (107k) | 739 | 17.92 |
| | SMp | 531 | **17.55** |
| Turkish | bpe (92k) | 402 | 13.53 |
| | Morfessor | 393 | 14.39 |
| | bpe (85k) | 474 | 13.04 |
| | SM | 426 | 11.07 |
| | bpe (60k) | 381 | 12.93 |
| | SMp | 432 | **10.67** |

Table 2: Pre-training results for English and Turkish GPTs with different segmentation methods. Vocabulary size (in parentheses) applies to all models in each box.

| | Segmenter | Size | steps (k) | PPL |
|---|---|---|---|---|
| Finnish | bpe (78k) | Base | 757 | 23.69 |
| | SMp | **Small** | 542 | **21.22** |
| | bpe | Small | 833 | 29.23 |
| | SMp | Base | 724 | 16.65 |
| Russian | bpe (170k) | Base | 660 | 24.97 |
| | SMp | **Small** | 651 | **23.39** |
| | bpe | Small | 1029 | 28.99 |
| | SMp | Base | 816 | 19.08 |

Table 3: Pre-training results for GPT language models of *different sizes* with BPE vs. SMp, compared in blue. Base models are the same as in Table 1.

terms of training time (RQ2).

## 4.2 Pre-Training with different model sizes

We next explore RQ4: model size. We train two more GPT models as above for Finnish and Russian—with morphological segmentation (SMp) and with BPE, but in smaller size, i.e., with a smaller number of parameters. In this experiment, we use models with the smallest vocabulary: 78k for Finnish and 170k for Russian.

Figure 3 shows the learning curves, and Table 3 shows the final perplexity at convergence for models of different sizes. We see that *smaller* LMs with morphological segmentation yield better perplexity compared to larger LMs with BPE segmentation. The smaller LMs have 132M parameters in Finnish, and 276M parameters in Russian, whereas the base LMs have 189M parameters in Finnish, 354M parameters in Russian—which is 43% and 28% bigger respectively.[6] This further confirms RQ4: morphological segmentation can help reduce the model size, and improve the sustainability of the models in the training and inference phases.

## 4.3 Fine-tuning for downstream tasks

We do not try to optimize the fine-tuning settings for the specific downstream tasks, since we *do not* aim for state-of-the-art performance. Rather our goal is to explore the impact of segmentation on the performance of LMs on downstream tasks. We run each downstream task experiment three times, and report the mean and standard deviation ($\sigma$) of the resulting scores. We evaluate classification task performance with 3 metrics: Accuracy, F1, and Matthews Correlation Coefficient (MCC).

**Topic classification (Finnish):** Table 4 shows in-domain and out-of-domain topic classification for both LM types. The performance of the LMs is overall quite close. The best performing model on the In-domain task is BERT with SMp segmentation, with an average accuracy of 92.6%. This is around 1% higher than its corresponding BPE-segmented LM. The best model on the Out-of-domain task is GPT with SM segmentation, with average accuracy of 79.3%. This is about 3% higher than its corresponding BPE-segmented LM.

All models achieve very good scores on In-domain data, and relatively reasonable scores on Out-of-domain. We apply the t-test to compare the morphologically-segmented LMs with their corresponding BPE-segmented LMs. Only BERT with SM shows a significantly worse performance than

---

[6]Details about the numbers of parameters for all models are given in Appendix A, Tables 10 and 11.

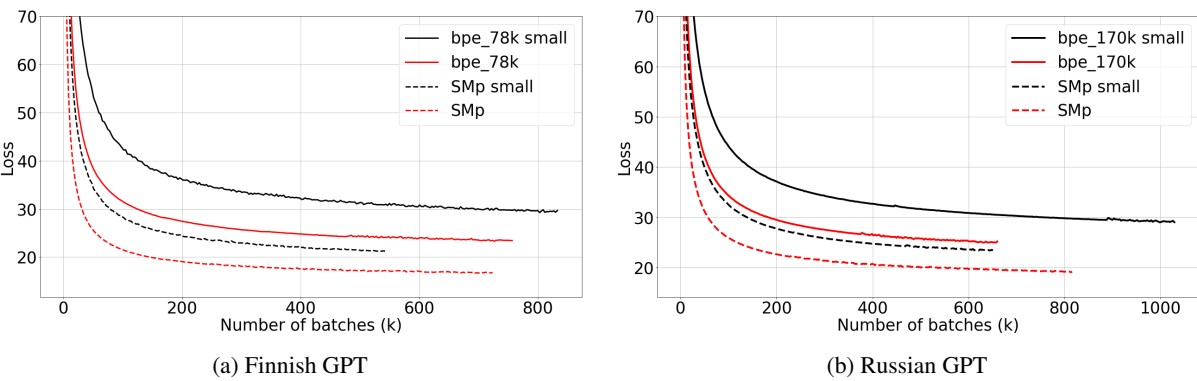

|     | (a) Finnish GPT | (b) Russian GPT |
| --- | --- | --- |

Figure 3: Learning curves for pre-training language models with different model sizes.

|     |     | In-domain (YLE corpus) | | | | | | Out-of-domain (Ylilauta corpus) | | | | | |
| --- | --- | --- | --- | --- | --- | --- | --- | --- | --- | --- | --- | --- | --- |
|     |     | Accuracy | | F1-measure | | MCC | | Accuracy | | F1-measure | | MCC | |
|     |     | Avg. | σ | Avg. | σ | Avg. | σ | Avg. | σ | Avg. | σ | Avg. | σ |
| GPT | bpe (233k) | **90.0** | 1.4 | 85.8 | 2.0 | 86.8 | 1.8 | 75.8 | 0.3 | 70.3 | 2.0 | 68.1 | 0.6 |
|     | morfessor | 89.4 | 1.0 | 85.0 | 1.4 | 85.7 | 1.6 | **78.4** | 2.4 | 72.2 | 3.4 | 71.2 | 3.3 |
|     | bpe (149k) | 87.9 | 1.2 | 84.2 | 1.2 | 84.1 | 1.6 | 77.2 | 2.2 | 70.8 | 3.3 | 69.7 | 3.5 |
|     | SM | **90.1** | 2.1 | 86.2 | 3.1 | 87.0 | 2.7 | **79.3** | 3.6 | 74.4 | 4.9 | 72.9 | 4.8 |
|     | bpe (78k) | 89.1 | 2.8 | 84.1 | 2.5 | 85.4 | 1.5 | 77.5 | 2.9 | 71.1 | 3.6 | 70.1 | 3.9 |
|     | SMp | **90.1** | 1.5 | 86.5 | 1.8 | 86.7 | 2.4 | **77.6** | 1.5 | 71.9 | 1.8 | 70.2 | 2.3 |
|     | SMp-sm | 88.4 | 2.8 | 84.8 | 3.5 | 84.9 | 3.6 | 75.5 | 2.5 | 69.4 | 4.3 | 67.3 | 3.4 |
|     | bpe-sm | 88.9 | 2.5 | 83.7 | 2.8 | 85.0 | 3.5 | 75.0 | 2.3 | 68.5 | 2.9 | 66.4 | 3.3 |
| BERT | bpe (233k) | **90.5** | 2.5 | 89.7 | 1.0 | 87.4 | 1.6 | 72.5 | 3.7 | 67.1 | 4.7 | 64.1 | 4.9 |
|     | morfessor | 90.2 | 1.1 | 89.6 | 2.8 | 87.0 | 3.5 | **75.4** | 1.5 | 69.5 | 1.8 | 67.5 | 2.2 |
|     | bpe (149k) | **91.8** | 1.3 | 91.3 | 1.0 | 89.3 | 1.6 | 75.4 | 3.7 | 70.7 | 4.3 | 67.5 | 4.8 |
|     | SM | 89.9 | 0.2 | 89.2 | 0.3 | 86.6 | 0.3 | **76.2** | 3.2 | 71.2 | 2.7 | 68.5 | 3.6 |
|     | bpe (78k) | 91.4 | 1.5 | 90.7 | 1.4 | 88.6 | 2.1 | **77.0** | 3.3 | 71.3 | 2.5 | 69.7 | 4.0 |
|     | SMp | **92.6** | 0.2 | 91.9 | 0.2 | 90.1 | 0.3 | 74.1 | 4.3 | 69.3 | 4.4 | 66.0 | 6.1 |

Table 4: Fine-tuning for Finnish topic classification.

BERT with BPE (149k), with p-value of 0.03. All other t-tests for all comparison pairs in all metrics do not show a significant difference in performance, with p-values all greater than 0.05. This suggests that the Finnish LMs with different segmentations perform comparably on topic classification after fine-tuning.[7] This includes smaller-size models (SMp-sm and BPE-sm). GPT with SMp-sm, which is smaller in size, is comparable with GPT with BPE (78k), which is of regular size. The minimum p-value of their t-test for all metrics is 0.2, while the maximum p-value is 0.41. This further suggests potential benefits for sustainability.

For reference, Virtanen et al. (2019) reach 90.6% accuracy on classifying YLE data, and 79.8% on

Ylilauta, with similar fine-tuning settings.

**Topic classification (Russian):** Table 5 shows the performance of each LM after fine-tuning. Overall, all models achieve very good performance on this task. The GPT model with SMp achieves the best overall performance, with an average accuracy of 91.4%, about 1% higher than the corresponding model with SM. As in Finnish, the small GPT model with SMp-sm is slightly better (1%) than base GPT with BPE (170k).

Among the BERT models, BERT with SM achieves the best performance overall with 87.3% accuracy. This is relatively 8% higher in accuracy and F1 than the corresponding model with BPE (516k), and about 11% higher in MCC. Compared to Finnish, the difference between models with morphological segmentation and BPE is larger (though not significantly) in accuracy and F1. To

---

[7]We use an unpaired t-test with unequal variance assumption. The null hypothesis is: a morphologically-segmented LM performs comparably to its BPE-segmented counterpart; the alternative hypothesis is: one LM outperforms the other.

| | | Accuracy | | $F_1$ | | MCC | |
|---|---|---|---|---|---|---|---|
| | | Avg. | $\sigma$ | Avg. | $\sigma$ | Avg. | $\sigma$ |
| GPT | bpe (488k) | **89.8** | 2.3 | 85.5 | 3.7 | 86.2 | 3.1 |
| | Morfessor | 89.3 | 0.9 | 84.6 | 0.4 | 85.8 | 1.1 |
| | bpe (516k) | **91.4** | 0.5 | 88.1 | 1.2 | 88.7 | 0.5 |
| | SM | 90.5 | 0.5 | 86.1 | 0.4 | 87.4 | 0.7 |
| | bpe (170k) | 88.5 | 1.8 | 83.5 | 2.2 | 84.5 | 2.5 |
| | SMp | **91.1** | 0.7 | 87.1 | 1.9 | 88.2 | 0.9 |
| | SMp-sm | 89.4 | 2.2 | 84.4 | 2.0 | 85.7 | 3.0 |
| | bpe-sm | 88.6 | 1.7 | 83.6 | 1.7 | 84.8 | 2.2 |
| BERT | bpe (488k) | **87.0** | 2.8 | 82.1 | 3.3 | 82.7 | 3.7 |
| | Morfessor | 82.7 | 4.3 | 77.6 | 4.3 | 77.0 | 5.9 |
| | bpe (516k) | 81.2 | 3.9 | 76.9 | 4.1 | 75.2 | 4.8 |
| | SM | **87.3** | 2.0 | 83.2 | 2.5 | 83.1 | 2.7 |
| | bpe (170k) | **87.1** | 1.9 | 82.5 | 1.9 | 82.8 | 2.4 |
| | SMp | 83.5 | 4.4 | 77.9 | 4.3 | 77.8 | 5.6 |

Table 5: Fine-tuning for Russian topic classification.

| | | Accuracy | | MCC | |
|---|---|---|---|---|---|
| | | Avg. | $\sigma$ | Avg. | $\sigma$ |
| GPT | bpe (488k) | **55.0** | 1.9 | 17.7 | **6.9** |
| | Morfessor | 52.5 | 1.9 | 15.1 | 5.5 |
| | bpe (516k) | 48.8 | 2.9 | 11.0 | **6.6** |
| | SM | **52.2** | 1.7 | 15.9 | 3.6 |
| | bpe (170k) | 52.7 | 6.5 | 6.2 | 8.4 |
| | SMp | 56.0 | 8.3 | 2.4 | 0.5 |
| | SMp-sm | 49.0 | 3.2 | 8.2 | 7.7 |
| | bpe-sm | **56.6** | 5.6 | 6.8 | **9.7** |
| BERT | bpe (488k) | 57.1 | 2.9 | 15.8 | 3.3 |
| | Morfessor | **60.5** | 6.8 | 10.6 | **9.3** |
| | bpe (516k) | **55.7** | 2.2 | 14.6 | **8.3** |
| | SM | 53.7 | 2.4 | 16.4 | 3.1 |
| | bpe (170k) | **54.8** | 5.0 | 11.7 | 2.1 |
| | SMp | 52.1 | 4.1 | 14.8 | **4.6** |

Table 7: Fine-tuning for Russian linguistic acceptability

| | | Accuracy | | MCC | |
|---|---|---|---|---|---|
| | | Avg. | $\sigma$ | Avg. | $\sigma$ |
| Finnish | bpe (233k) | **95.0** | 0.1 | 94.0 | 0.1 |
| | Morfessor | 94.0 | 0.1 | 92.9 | 0.0 |
| | bpe (149k) | 95.0 | 0.0 | 94.0 | 0.0 |
| | SM | **95.4** | 0.0 | 94.5 | 0.0 |
| | bpe (78k) | 95.3 | 0.1 | 94.5 | 0.1 |
| | SMp | **95.4** | 0.0 | 94.5 | 0.0 |
| Russian | bpe (488k) | **97.6** | 0.0 | 97.2 | 0.0 |
| | Morfessor | 97.5 | 0.0 | 97.1 | 0.0 |
| | bpe (516k) | **97.7** | 0.0 | 97.3 | 0.0 |
| | SM | 97.6 | 0.0 | 97.2 | 0.0 |
| | bpe (170k) | **98.0** | 0.0 | 97.7 | 0.0 |
| | SMp | 97.9 | 0.0 | 97.5 | 0.0 |

Table 6: Fine-tuning for Finnish PoS tagging.

confirm this, we apply the t-test, comparing the morphological BERTs with their corresponding BPE-segmented BERTs, for accuracy and F1.

The average t-test p-values on morphological BERT vs. BPE BERT are 0.1 and 0.09, for accuracy and F1, respectively; the p-values for the t-tests in Finnish are 0.45 and 0.42. This indicates more significant difference between morphological BERT vs. BPE BERT in Russian than in Finnish. But the performance of morphological BERTs is still comparable with BPE BERTs in terms of accuracy and F1, as their p-values are over 0.05.

Public leaderboards show state-of-the-art accuracy of about 96% on this task.[8]

**PoS tagging:** Table 6 shows the results for Finnish and Russian. All models achieve very good accuracy and MCC. The BERT models with

SM and SMp achieve the best performance, but this is only slightly better than the performance of BERT with BPE (149k) and BPE (78k), while BERT with Morfessor is worse by 1%. In Russian, LMs with morphological segmentation have generally the same performance as LMs with BPE (less than 0.1% difference). Overall, performance is very close on this downstream task.

For comparison, Virtanen et al. (2019) achieve 98.23% accuracy for Finnish with fine-tuning on the same dataset; others achieve 97.8% accuracy for Russian on this task.[9]

**Linguistic acceptability:** Table 7 shows the results on Russian. This is a very difficult task, with accuracy only around 50–60%. Overall, the BERT model with Morfessor segmentation achieves the best accuracy, not significantly better than the corresponding BPE, which p-value of their t-test is 0.25. BERT and GPT achieve a relatively close performance. The small GPT model with SMp segmentation performs worse than regular GPT with BPE (170k), in relative terms by 7% on accuracy, but better on MCC, by 32%.

For comparison, GPT-3 in (Mikhailov et al., 2022) reaches 55.8% accuracy on this task, while BERT reaches 75.9% accuracy.

**Paraphrasing:** Table 8 shows the results of experiments on paraphrase generation, another extremely challenging task, even for the human. The evaluation metric chrF++, from machine translation (Popović, 2017, 2015), uses the F-score statistic for character n-gram and word n-gram matches.

[8]Kaggle leaderboard, text classification (RU).

[9]XLM-RoBERTa base, UD POS tagging: Russian

| | | *average* chrF++ | $\sigma$ |
|---|---|---|---|
| Finnish | bpe (233k) | 28.9 | 0.5 |
| | Morfessor | **29.6** | 0.5 |
| | bpe (149k) | 30.8 | 1.2 |
| | SM | **31.5** | 0.5 |
| | bpe (78k) | **32.8** | 0.6 |
| | SMp | 31.9 | 1.4 |
| | SMp-sm | 27.2 | 0.3 |
| | bpe-sm | 28.2 | 2.1 |
| Russian | bpe (488k) | 24.0 | 0.1 |
| | Morfessor | **25.8** | 0.7 |
| | bpe (516k) | **26.2** | 0.2 |
| | SM | 26.0 | 0.6 |
| | bpe (170k) | 22.6 | 0.8 |
| | SMp | **28.9** | 0.0 |
| | SMp-sm | 22.2 | 0.8 |
| | bpe-sm | 22.6 | 0.3 |

Table 8: Fine-tuning for paraphrase generation

The Finnish models and most of the Russian models show fairly similar performance. The performance of the Russian models with SMp segmentation is relatively better than the models with BPE segmentation by 28%. The small Russian models with SMp segmentation is 2% below the regular model with BPE (170k), while the small Finnish model with SMp-sm segmentation is 21% below the regular model with BPE (78k), in relative terms.

## 5 Conclusions and future work

We have explored the impact of intelligent segmentation algorithms on the performance of language models. We experiment with four languages: Finnish, Russian, Turkish, and English, with an in-depth investigation of the first two.[10] We train two language models—GPT and BERT—and compare the statistical segmentation algorithm, BPE, with two morphological segmentation algorithms: Morfessor and StateMorph.

We aim to show that LMs trained on a vocabulary based on morphological information are better than LMs trained on "naïve" sub-word segments produced by BPE. Although BPE does not explicitly model morphology, it will inevitably stumble into discovering *some* morphemes as well—because many morphemes also happen to be frequent subwords. This makes BPE a tough baseline to beat.

We explore four research questions: does morphological segmentation help LMs— 1: reach lower perplexity; 2: learn and converge faster; 3: perform at least as well on downstream tasks; 4: perform at least as well with smaller model size.

We show that LMs trained with morphological segmentation reach much lower perplexity (except Morfessor, which has the largest vocabulary) than LMs trained with BPE (RQ1). We also show that LMs trained with morphological segmentation converge faster than LMs trained with BPE (RQ2).

We evaluate the performance of language models on several downstream tasks (RQ3). The results show that the performance of LMs with morphological segmentation (including smaller LMs) is comparable to models with BPE. While performance on the topic classification tasks is quite convincing, we meet several challeges with other downstream tasks. The tasks are so high-level and so difficult, and the baseline performance is so low, that the *gains* from "smarter" segmentation may not be easy to demonstrate directly. The languages we work with have a paucity of "standard" datasets for downstream tasks with sufficient labeled data. In the future, we will investigate more languages and more downstream tasks. However, the theoretical results from RQ1, 2 and 4 are convincing.

To investigate the impact of segmentation on model size (RQ4), we pre-train a smaller version of each LM with StateMorph, for each language. We show that—for a fixed vocabulary size—small LMs with StateMorph segmentation have lower perplexity than regular-sized LMs with BPE segmentation.

This suggests that morphological segmentation can reduce the size of language models, and improve the *sustainability* of LMs (RQ4). Sustainability is impacted by RQ2, but even more so by RQ4, since RQ2 affects only the training, whereas RQ4 affects training and inference.

In future work, we plan to experiment with more morphological segmentation algorithms, a broader range of languages, and more types of language models, such as Transformer-XL (Dai et al., 2019) and XLNet (Yang et al., 2019). As a final point, the smarter segmentations that we have tested have much room for improvement—e.g., we can expect that supervised or rule-based morphological segmentation will be still better than the unsupervised segmentation that we have tested so far.

---

[10] The languages are chosen as representatives of their respective sub-families—Finno-Ugric and Slavic—which have very rich morphology, both verbal and nominal, certainly among the richest among the European languages.

## Acknowledgements

This research was supported in part by BusinessFinland (Grant 42560/31/2020), and by a grant from the Helsinki Institute for Information Technology (HIIT). We are grateful for assistance from Javad Nouri.

## Limitations

We acknowledge that this work has several limitations. First, we use only two language model architectures, GPT and BERT. Second, we use only two languages, Finnish and Russian, for each architecture. Third, we acknowledge our selection of segmentation algorithms is limited; other segmentation algorithms exist, both supervised or unsupervised. We plan to investigate other languages and the impact of different segmentation algorithms as well as different LM architectures and settings in future work.

We also acknowledge that the size of the lexicon is a factor which impacts the performance of the language model. Due to limited computational resources, we experimented with a limited choice of lexicon sizes, where some of them may not be optimal. We plan to investigate further the effects of lexicon size. Lastly, we explored only a limited number of downstream tasks, which may not reveal the complete picture about the performance of a language model.

## Ethics Statement

This data used in this work is mostly open data, or data used with explicit permission from its publisher. We do not use any private data, nor any data that is not allowed to be used for research purposes.

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

## A  Pre-Training details

|  | Base | Small |
|---|---|---|
| Transformer blocks (L) | 12 | 8 |
| Self-attention heads (A) | 12 | 8 |
| Hidden size (H) | 768 | 512 |
| Feed-forward/filter | 1024 | 512 |

Table 9: Hyperparameters for GPT/BERT models.

We pre-train all language models with the same settings for the optimizer. We apply AdamW optimizer, mostly with the default parameters from Torch, except we use (0.9, 0.998) for the beta values. We use batch size of 100 for most Russian and Finnish models, except Russian GPT with Morfessor, SM, and BPE with the corresponding vocabulary size. We are not able to fit Russian models and data into the GPU memory during training, therefore we compensate by training with batch size 50, and accumulate gradient for 2 steps to achieve the same effect as batch size of 100. We use batch size 300 for all English and Turkish GPTs to speed up the pre-training process. We use the same learning rate (5e-5) for all models, except for the Finnish GPT models, which uses a larger learning rate (1e-4). We use a different number of validation steps depending on model type: 300 for the GPT models, and 1200 for BERT models. We use a larger number of validation steps, because the random masking mechanism decrease the actual tokens for validation, as compared to GPT models. We use EarlyStop with a patience of 10 and $\delta = 10^{-5}$.

Table 9 shows the hyperparameters used for pre-training GPT and BERT models. Tables 10 and 11 show the vocabulary size of different segmentations, and their corresponding overall number of parameters, when pre-training Finnish and Russian language models, respectively.

We pre-train in two stages for all of the Finnish and Russian GPT models. We first pre-train each

| Segmenter | Size | Voc (K) | #Param (M) |
|---|---|---|---|
| Morfessor | Base | 233 | 467 |
| bpe | Base | 233 | 473 |
| SM | Base | 149 | 315 |
| bpe | Base | 149 | 315 |
| SMp | Base | 78 | 188 |
| SMp | Small | 78 | 132 |
| bpe | Base | 78 | 189 |
| bpe | Small | 78 | 134 |

Table 10: Segmentation of Finnish data. Corresponding vocabulary sizes (thousands of tokens), and model size (millions of parameters)

| Segmenter | Size | Voc (K) | #Param (M) |
|---|---|---|---|
| Morfessor | Base | 487 | 921 |
| bpe | Base | 488 | 923 |
| SM | Base | 518 | 979 |
| bpe | Base | 516 | 974 |
| SMp | Base | 171 | 355 |
| SMp | Small | 171 | 276 |
| bpe | Base | 170 | 354 |
| bpe | Small | 170 | 275 |

Table 11: Segmentation of Russian data. Corresponding vocabulary sizes (thousands of tokens), and model size (millions of parameters)

model on a GPU cluster for the first 36 hours. Each node in the cluster is equipped with 4 Nvidia A100-40G GPUs and two AMD Rome 7H12 CPUs with 64 cores each. We use all GPUs and 64 cores for each job. We then continue the pre-training on a bigger cluster, until the models converge, where each cluster node is equipped with 4 AMD MI250x GPU modules. Each GPU module has a AMD EPYC "Trento" CPU and two GPU dies with 64GB of HBM2 memory, which makes 8 GPUs overall in one node. We request the same number of GPUs and CPU cores for each model, as in the jobs we run on the first cluster. For BERT models as well as English and Turkish GPT models, we pre-train only on the second cluster, with the same settings as for the GPT models.

### A.1  Segmentation and capitalization

A technical point of difference between BPE and the other segmenters: BPE distinguishes tokens appearing word-initially vs. elsewhere, with a special symbol "_", to designate whitespace, for example, "_ba·king". Morfessor and StateMorph do not distinguish in their output whether a morpheme appears

initially or medially.[11]

Therefore we perform segmentation as follows: pre-segment all words in lower case, then convert the tokens back to the original case, mark all tokens appearing word-initially with "_" (as BPE does), and collect the exact vocabulary for pre-training language models.

## B Fine-tuning details

We follow a similar training process as pre-training. We conduct all fine-tuning tasks on the second cluster, which is also used in pre-training, with the same resources as in pre-training. We use the same optimizer (AdamW), and same optimizer parameters as in pre-training. We apply the same learning rate (5e-5), and the same batch size of 50; we use 50 rather than 100 in pre-training, so that all models can be fine-tuned uniformly. We uniformly accumulate gradient for 5 steps.

---

[11]They do model the beginning or end of a word when learning to segment.