# OpenReview forum: "Effects of sub-word segmentation on performance of transformer language models"
_EMNLP/2023/Conference — EMNLP 2023 Main_

### Official Review · Reviewer_knhq · 2023-07-26

**Soundness:** 4

**Excitement:**

4: Strong: This paper deepens the understanding of some phenomenon or lowers the barriers to an existing research direction.

**Paper Topic And Main Contributions:**

This paper compares the performance of two unsupervised sub-segments Morfessor and StateMorph with BPE on training language models and downstream tasks.

**Questions For The Authors:**

Line 257, why doesn't it include the next sentence prediction? What does it have to do with instance size 256? This setting may have an effect on the experimental results.

Line 394, the performance of each segmentation method on fine-tuning is also important.

**Reasons To Accept:**

The authors demonstrate that Morfessor and StateMorph are more effective than BPE for small model size settings. This is very useful for practical applications.

**Reasons To Reject:**

The authors study only two morphologically rich languages—Finnish and Russian. It is unclear whether the results apply to other languages. This would prevent the widespread use of these two alternative segmentation methods.

**Reproducibility:**

4: Could mostly reproduce the results, but there may be some variation because of sample variance or minor variations in their interpretation of the protocol or method.

**Reviewer Confidence:**

4: Quite sure. I tried to check the important points carefully. It's unlikely, though conceivable, that I missed something that should affect my ratings.

---

> ### Author Rebuttal · Authors · 2023-08-28
>
> Thank you for your review.
>
> _The authors study only two morphologically rich languages—Finnish and Russian. It is unclear whether the results apply to other languages. This would prevent the widespread use of these two alternative segmentation methods._
>
> (Similarly to our answer to review VFFv —)
> Ideally, this would be tested on many languages. We present work on two languages, so far, because:
> * We are a small academic team, not a large corporation, and we have limited computing resources.
> * Even for two languages, we already had to train and run hundreds of models.
> * We follow the zero-one-infinity rule: If the phenomenon is observed in two or more highly distinct cases, it is probably of significance (though not guaranteed).
> * For many (other) morphologically rich languages, resources for training unsupervised models also are limited; resources for training downstream tasks are even more limited.
> * Our languages are quite representative: they have very different morphological structures — one agglutinative (FIN), and one fusional (RUS).
> * In fact, we are in the process of testing our hypotheses also on English and Turkish. If the paper is accepted, we will include those results in the additional space provided.
>
> Overall, we do not claim that the study is comprehensive in regard to the number of languages as well as segmentation algorithms. Our claim is this is an important problem to investigate further, with more languages and tasks.
>
> The goal of our paper is:
>  * to present our investigations so far,
>  * draw attention to this important issue, and
>  * to open the discussion and further exploration.
> It is quite expensive (in computational resources) to add each additional language.
>
>
> *** Questions For The Authors:
>
> **** _Line 257, why doesn't it include the next sentence prediction?_
>
> We are concerned that with the _smaller instance size_ — to which we are constrained by our GPU resources — make it place limitations on the next-sentence prediction task.
> In future work (when we have more computational resources) we agree, one should test with training on this task, as well.  However, when training with this additional task, we can expect that _both_ models will perform stronger as a result — both BPE _and_ morphologically aware segmentation.
>
> We should note that:
> 1. *Our* goal is not to beat state-of-the-art results on BERT/GPT
> 2. For our experiment and our goals:
>     - we are not aiming to compare our model(s) with models trained in "stronger" settings.
>     - we want to make sure that the models are A. comparable *among themselves*, B. trainable in reasonable time.
>         - thus we trained them on somewhat simpler task. However, among themselves these models are comparable.
>         - the next-sentence prediction task is not necessary _for comparability_.
>     - can be explored in future work.
>
>
> **** _What does it have to do with instance size 256? This setting may have an effect on the experimental results._
>
> In line 257: _"Due to limited computational resources, we used a smaller instance size---256, half of what is typically used for BERT."_ ⇒  We did not want to "waste" model parameters on a harder task.  Rather we focus only on the segmentation, and the effect of segmentation on the end result.
>
> We aimed for a more pure experiment - attainable by/available to more researchers: smaller model size.
>
> In future work, we (and others) can repeat this on bigger models, with more computational resources.
>
>
> **** _Line 394, the performance of each segmentation method on fine-tuning is also important._
>
> We do not aim for optimal hyper-parameters for the different segmentation methods, since that would make them not comparable among themselves.  We fix the hyper-parameters, and compare the performance on downstream tasks _across the segmentation models_.

---

### Official Review · Reviewer_HPcL · 2023-08-01

**Soundness:** 4

**Excitement:**

4: Strong: This paper deepens the understanding of some phenomenon or lowers the barriers to an existing research direction.

**Missing References:**

not aware of missing references.

**Paper Topic And Main Contributions:**

The article discusses the effects of different sub-word segmentation algorithms to transformer-based language models.

The authors compare three segmentation algorithms, namely BPE, Morfessor and StateMorph, using  two morphologically complex languages, Finnish and Russian.



**Questions For The Authors:**

One question would be why the authors do not refer to the sub-word segmentation algorithm included in Sentencepiece (Kudo et al.), whcih appears to be widely used in recent years.

It is not especially clear which version of the Morfessor is used. Would it be possible to estimate what the more advanced methods would confer in terms of improvement (as far as I understand, StateMorph is better than the Morfessor variant used)?

Line 255: "Due to limited computational resources, we used 256
a smaller instance size" - could the authors elaborate on this? I suppose they could refer to the material available in the Appendix.

Line 286: "We set the lexicon size for BPE
287 manually—to match the lexicon size produced by
288 the morphological segmentation algorithms. Sizes
289 of the resulting lexicons are shown below." - Did the two morphological-based  algorithms (Morfessor/StateMorph) lead to similar lexicon sizes?  What if BPE had been used to determine the lexicon size for all 3 algorithms; in such a case would the results have changed substantially to make BPE a more competitive method?

Line 367: "the number of steps to reach conver- 367
gence is higher for most models with BPE, sev- 368
eral times higher for some." - indicate which entries in the Table this statement corresponds to, so as to aid readability.

Line 415: "The overall variation is less than 5%, 415
which suggests that Finnish LMs with different seg- 416
mentations perform comparably on topic classifi- 417
cation after fine-tuning." - have the authors checked if this variation of less than 5% is not statistically significant?

Line 440: "the difference between models with mor441
phological segmentation and BPE is larger (though
442 not significantly) in terms of accuracy and F1." - it would be useful if the authors could discuss the question of statistically significant (or non-significant) differences.

It  would be useful to determine the acronym MCC (I assume it corresponds to the Matthews correlation coefficient).

A minor point: "Finnish and Russian. These languages are chosen
493 as representatives of their respective sub-families—
494 Slavic and Finno-Ugric" - change the sequence of the last two terms, at line 494 to agree with the order in line 492.



**Reasons To Accept:**

The article contains extensive experiments on comparing the different sub-word segmentation algorithms.

The description of the problem is clear and a set of 4 Research Questions (RQ1 to RQ4)  are set early in the manuscript.

Also, in each of the downstream tasks discussed in the experimental section,  the authors provide details on the performance of dedicated systems to compare these to their own experiments.

To my understanding the experiments are thorough and well defined.

Each experimental run  is carried out 3 times which is commendable.

Also, the sustainability aspect has been looked into by investigating smaller models.

The linguistic quality is good as well as the method of exposition of material. Also, the system parameters are provided in the Appendix, to aid reproducibility.


**Reasons To Reject:**

There are a few weaknesses in the manuscript, but I believe this is a sound and interesting paper that would be of use to the EMNLP community and probably also beyond.

**Reproducibility:**

4: Could mostly reproduce the results, but there may be some variation because of sample variance or minor variations in their interpretation of the protocol or method.

**Reviewer Confidence:**

4: Quite sure. I tried to check the important points carefully. It's unlikely, though conceivable, that I missed something that should affect my ratings.

**Typos Grammar Style And Presentation Improvements:**

Line 249: "answer RQ1: does a smarter 249
segmentation help the LMs learn faster." - should there be a question mark at the end?

Line 483: "The small Russian models
484 with SMp segmentation is 2% below the regular
485 model with BPE (170k)," - check agreement of verb to subject.

---

> ### Author Rebuttal · Authors · 2023-08-28
>
> Thank you for your review.  Some responses to your questions:
>
> **** _One question would be why the authors do not refer to the sub-word segmentation algorithm included in Sentencepiece (Kudo et al.), which appears to be widely used in recent years._
>
> In our implementation, we do indeed use Google's latest version of the Sentencepiece library, which implements BPE.  We did not mention it in the paper, but we should (and will do so, if the paper is accepted).
>
> **** _It is not especially clear which version of the Morfessor is used.  Would it be possible to estimate what the more advanced methods would confer in terms of improvement (as far as I understand, StateMorph is better than the Morfessor variant used)?_
>
> We have reported the use of Morfessor 2.0 implementation (Virpioja et al., 2013) of the baseline method.  You are correct, the experiments show that StateMorph is better than this Morfessor variant.
> We should indeed test the more recent Morfessor variants, and we could expect more competitive performance.  Some of the later variants would allow us to control the size of the segment lexicon — the fixed lexicon size is a limitation of the baseline method.
> The standard distribution offers Morfessor version 2.0.6.
>
> Therefore, yes, we would expect improvements from later and more sophisticated variants of Morfessor.  We are in fact testing Morfessor EM+Prune (https://aclanthology.org/2020.lrec-1.486/). We have found version 2.0.7 from one of the authors of this paper, and are comparing the performance with the other algorithms (to be included in the final paper, if accepted).
>
>
> **** _Line 255: "Due to limited computational resources, we used a smaller instance size" - could the authors elaborate on this? I suppose they could refer to the material available in the Appendix._
>
> You are correct, Appendix (A) mentions this, in line 777; at line 798 we describe our resources.  Unfortunately, instances of bigger size do notf fit into our GPUs.  We should (and will) make this more clear in the main text, Section 3.1.
>
>
> **** _Line 286: "We set the lexicon size for BPE manually—to match the lexicon size produced by the morphological segmentation algorithms. Sizes of the resulting lexicons are shown below." - Did the two morphological-based algorithms (Morfessor/StateMorph) lead to similar lexicon sizes? What if BPE had been used to determine the lexicon size for all 3 algorithms; in such a case would the results have changed substantially to make BPE a more competitive method?_
>
> Yes, this is a very important delicate point.
> The lexicon size can be controlled for BPE.  We could indeed go in the opposite direction and allow BPE to determine the lexicon size X that is somehow "optimal" — then StateMorph could be pruned to the same size X.  Also, Morfessor EM+Prune could be restricted to size X, and then we could compare the three models in that setting.
>
> We should emphasize more clearly that at _every lexicon size_ that we explored, BPE always turned out to be inferior.  Therefore, given all the results collected so far, we would expect BPE to be inferior for other settings of the lexicon size.
>
> Finding the "optimal" size for BPE — in terms of perplexity — would require an extensive search: as the lexicon size decreases, we expect the perplexity to improve, until some critical point, after which it will again grow.
>
> To answer the question "where is BPE's *OWN* optimum vocabuary size", we would need to determine that for each language, and for each dataset.
>
> We will also provide a more clear explanation that our morfessor variant is *NOT* customizable w.r.t. the vocabuary size.  Other variants are, and will be tested.
>
>
> **** _Line 367: "the number of steps to reach convergence is higher for most models with BPE, several times higher for some." - indicate which entries in the Table this statement corresponds to, so as to aid readability._
>
> Yes, this should (and will) be clarified — it refers to the first column "Steps (k)" in Table 1.
>
> **** _Line 415: "The overall variation is less than 5%, which suggests that Finnish LMs with different segmentations perform comparably on topic classification after fine-tuning." - have the authors checked if this variation of less than 5% is not statistically significant?_
>
> Yes, the statement in the paper may be a bit confusing.  We should clarify it to state:
>
> "For the In-Domain measures (Accuracy, F1, MCC), for 3 out of the 6 tested conditions, the morphological segmentation is better than BPE.  For the Out-Of-Domain measures, for 5 out of the 6 tested conditions, morphological segmentation performs better (by within 5 percentage points). From this we conclude that the models — morphological and BPE — perform comparably overall."
>
> We will also check and report the statistical significance, as you suggest, though that was not the intent in the original statement.  But it's a good idea.
>
>
> **** _Line 440: "the difference between models with morphological segmentation and BPE is larger (though not significantly) in terms of accuracy and F1." - it would be useful if the authors could discuss the question of statistically significant (or non-significant) differences._
>
> These statements are making general comparisons, to say that the performance is comparable.
> In the final paper, we will provide more careful calculations of statistical significance.
>
>
> **** _It would be useful to determine the acronym MCC (I assume it corresponds to the Matthews correlation coefficient)._
>
> Yes, we will add a clarification that this MCC does stand for that.
>
>
> **** _A minor point: "Finnish and Russian. These languages are chosen 493 as representatives of their respective sub-families— Slavic and Finno-Ugric" - change the sequence of the last two terms, at line 494 to agree with the order in line 492._
>
> Absolutely  :)
>
> (Thank you for spotting the unfortunate reversal.)
>
> Thank you also for spotting and pointing out the _typos_, we will of course fix those!  Very helpful.

---

### Official Review · Reviewer_VFFv · 2023-08-05

**Soundness:** 3

**Excitement:**

3: Ambivalent: It has merits (e.g., it reports state-of-the-art results, the idea is nice), but there are key weaknesses (e.g., it describes incremental work), and it can significantly benefit from another round of revision. However, I won't object to accepting it if my co-reviewers champion it.

**Paper Topic And Main Contributions:**

The paper explores the impact of segmentation algorithms on language models. Experiments were conducted on two morphologically rich languages, Finnish and Russian. They tested GPT and BERT models trained on different segmentation algorithms, morphological (Morfessor or StateMorph) against a statistical one (BPE). Results shows that language models trained with a morphological segmenter converge faster than those trained with BPE. Besides that, smaller models trained with morphological segmenters are comparable to larger ones trained with BPE.

**Reasons To Accept:**

Sub-word segmentation algorithms are important in language modeling and more studies are needed.

**Reasons To Reject:**

A comprehensive study would cover more languages and algorithms.

**Reproducibility:**

4: Could mostly reproduce the results, but there may be some variation because of sample variance or minor variations in their interpretation of the protocol or method.

**Reviewer Confidence:**

3: Pretty sure, but there's a chance I missed something. Although I have a good feel for this area in general, I did not carefully check the paper's details, e.g., the math, experimental design, or novelty.

---

> ### Author Rebuttal · Authors · 2023-08-28
>
> Thank you for your review.
>
> _A comprehensive study would cover more languages and algorithms._
>
> True, ideally this would be tested on many languages.
> We present work on two languages, so far, because:
>   * We are a small academic team, not a large corporation, and we have limited compute resources.
>   * Even for two languages, we already had to train and run *hundreds of models*.
>   * We follow the _zero-one-infinity rule_: If the phenomenon is observed in two or more highly distinct cases, it is probably of significance (though not guaranteed).
>   * For many (other) morphologically rich languages, resources for training unsupervised models also are limited; resources for training downstream tasks are even more limited.
>   * Our languages are quite representative: they have very different morphological structure — one agglutinative (FIN), one fusional (RUS).
>   * In fact, we are in the process of testing our hypotheses also on English and Turkish. If the paper is accepted, we will include those results in the additional space provided.
>
> _Cover more algorithms:_
>
> We would be happy to test more segmentation algorithms.  Could you please provide an indication which algorithms we should test?
>
> What our paper presents is:
>  * 99.9% of all work involving language models uses BPE.
>  * Our initial experiments (on two quite different languages) find that BPE is sub-optimal:
>      - It causes the model to waste parameters to compensate for the naive segmentation.
>      - Based on the initial experiments, we observe that morphologically aware models could be at least 30-40% smaller (!)
>      - (a deeper exploration is needed, which may show even bigger gains.)
>  * This means that the standard approach produces models that are slower *and* burn more electricity.
>
> We do not claim that the study is comprehensive in regard to the number of languages.  Our claim is this is an _important problem_ to investigate further, with more languages and tasks.

---

### Meta-Review · Area_Chair_DXwi · 2023-09-29

**Recommendation:** 4

**Metareview:**

Paper presents a detailed study and analysis of different sub-word segmentation algorithms and their impact on convergence, generalization and efficiency. Reviewers unanimously found the paper to be sound and exciting with few editorial and experiemental design issues.

---

### Decision · Program_Chairs · 2023-10-07

**Decision:**

Accept-Main

**Comment:**

Paper presents a detailed study and analysis of different sub-word segmentation algorithms and their impact on convergence, generalization and efficiency. Reviewers unanimously found the paper to be sound and exciting with few editorial and experiemental design issues.